# microRNAs: Key Players in Plant Response to Metal Toxicity

**DOI:** 10.3390/ijms23158642

**Published:** 2022-08-03

**Authors:** Ying Yang, Jiu Huang, Qiumin Sun, Jingqi Wang, Lichao Huang, Siyi Fu, Sini Qin, Xiaoting Xie, Sisi Ge, Xiang Li, Zhuo Cheng, Xiaofei Wang, Houming Chen, Bingsong Zheng, Yi He

**Affiliations:** 1State Key Laboratory of Subtropical Silviculture, Zhejiang A&F University, Hangzhou 311300, China; sky_yangying@163.com (Y.Y.); sunqiumin99@163.com (Q.S.); jingqiwang1999@163.com (J.W.); huanglichao@zafu.edu.cn (L.H.); fusiyis@163.com (S.F.); qinsini98@163.com (S.Q.); xiexiaoting@stu.zafu.edu.cn (X.X.); vin@stu.zafu.edu.cn (S.G.); xxkk0209@stu.zafu.edu.cn (X.L.); chengzhuo626717@163.com (Z.C.); xfwang@zafu.edu.cn (X.W.); 2School of Environment Science and Spatial Informaftics, China University of Mining and Technology, Xuzhou 221116, China; jhuang@cumt.edu.cn; 3Max Planck Institute for Biology, Max Planck Ring 5, 72076 Tübingen, Germany; houming.chen@tuebingen.mpg.de

**Keywords:** heavy metals, miRNA, toxicity response, transcription factor, response

## Abstract

Environmental metal pollution is a common problem threatening sustainable and safe crop production. Heavy metals (HMs) cause toxicity by targeting key molecules and life processes in plant cells. Plants counteract excess metals in the environment by enhancing defense responses, such as metal chelation, isolation to vacuoles, regulating metal intake through transporters, and strengthening antioxidant mechanisms. In recent years, microRNAs (miRNAs), as a small non-coding RNA, have become the central regulator of a variety of abiotic stresses, including HMs. With the introduction of the latest technologies such as next-generation sequencing (NGS), more and more miRNAs have been widely recognized in several plants due to their diverse roles. Metal-regulated miRNAs and their target genes are part of a complex regulatory network. Known miRNAs coordinate plant responses to metal stress through antioxidant functions, root growth, hormone signals, transcription factors (TF), and metal transporters. This article reviews the research progress of miRNAs in the stress response of plants to the accumulation of HMs, such as Cu, Cd, Hg, Cr, and Al, and the toxicity of heavy metal ions.

## 1. Introduction

The rapid growth of industrialization and the large-scale use of chemical fertilizers and pesticides have led to the continuous increase in heavy metal content in the soil. In plants, copper (Cu), zinc (Zn), iron (Fe), and manganese (Mn) are trace elements necessary for plant development and growth, but excessive accumulation can cause cell damage [1]. Some other metals such as cadmium (Cd), chromium (Cr), lead (Pb), aluminum (Al), arsenic (As), and mercury (Hg), as non-essential elements, are toxic even at low concentrations [2]. Studies have reported that heavy metal stress can inhibit activity of antioxidant enzyme systems and differential expression of a large number of proteins in plants, weakening photosynthesis, accompanied by a series of phenotypes such as suppressed root development and leaf senescence or even necrosis [3]. In order to avoid the destructive consequences of heavy metal toxicity, plants have developed corresponding coping mechanisms to resist heavy metal stress [3]. Plants can reduce heavy metal concentrations in their bodies by limiting the uptake of heavy metal ions and stimulating metal efflux, as well as by complexation of metal ligands such as glutathione (GSH), metallothionein (MT), and phytochelatins (PCs) [4,5,6]. In addition, the antioxidant defense mechanism will also be activated to reduce elevated reactive oxygen species (ROS) levels, thereby reducing oxidative damage [7,8]. A large number of studies have shown that gene expression plays a very important role in regulating the tolerance of HMs or individually regulating various stress response genes to form a gene network. Some functional group genes encode metabolites such as amines, alcohols, and sugars, which also play a crucial role in heavy metal stress tolerance [9,10].

With the discovery of small RNAs, there is growing interest in the importance of post-transcriptional gene regulation of microRNAs (miRNAs) in plant development and responses to environmental stresses. miRNAs are an extensive class of small noncoding (19~24 nt) RNAs molecules [11]. In plants, mature miRNAs are produced through a multistep process including the transcription, precursor processing, methylation, and assembly of miRNA-induced silencing complex (miRISC) [12]. Then, the mature miRNAs lead the RISC to target the complementary mRNAs, which play a vital post-transcriptional regulatory role in gene expression by target mRNA cleavage or translational inhibition [13]. In recent years, a crescendo of miRNA studies has demonstrated that miRNAs play important roles in tissue development and differentiation, phytohormones signaling, secondary metabolite production, and biotic and abiotic stress [14,15]. miRNAs affect multiple processes of plant growth, development, and response to stress by up-regulating or down-regulating their expression [16,17]. miR165/166 and miR394 have been shown to be involved in shoot apical meristem (SAM) development, including the direct post-transcriptional regulation of key SAM-related genes, which in turn maintains SAM development [18,19]. In addition, studies have shown that miR319 has a conserved regulatory function in leaf development. For example, ectopic upregulation of miR319 resulted in dramatic changes in tomato leaf size and shape [20]. *CUC1* and *CUC2* mRNAs accumulate in the axil of leaf primordia and play a key role in the establishment of axillary bud meristems. The regulatory mechanism of miR164-CUC1/CUC2 may be related to LAS-mediated initiation of axillary bud meristems [21]. In addition, miRNAs play important roles in plant root development. miR160 acts as a key controller and cleaves *ARF10* and *ARF16* transcripts during plant root cap formation [22]. In contrast, miR167 plays an active role in adventitious root formation, while miR156 and miR172 are well-studied miRNAs involved in floral control [23,24]. Overexpression of miR172 can promote flowering time in both monocotyledonous and dicotyledonous plants. In contrast, the expression level of miR156 decreased gradually from sowing to flowering, while upregulation of miR156 resulted in delayed flowering transition. These findings suggest that miRNAs play regulatory roles in different developmental transitions by mediating specific signaling pathways. In different models and crop plants (such as Arabidopsis, wheat, rice, maize, and barley), miRNAs regulate gene expression during different stress responses (drought, heat, salinity, cold, nutrition, and pathogens). In addition to plant growth and development, the role of conserved miRNA target modules is also critical for conferring stress tolerance through integration into metabolic pathways [25]. Studies have confirmed that miR160-ARF, miR156-SPL, miR159-MYB33, miR164-NAC, miR172-AP2, miR394-LCR, miR396-GRF, and miR398-CSD modules play important regulatory roles in different stress environments to mitigate the effects of adverse reactions [12]. For example, in Arabidopsis, increased expression of miR398 enhances plant heat tolerance by negatively regulating the expression of its targets *CSD1*, *CSD2*, and the copper chaperone (*CCD*) of CSD [26]. The highly conserved miR394-LCR module is involved in plant responses to cold stress [27]. The expression of auxin-responsive factors ARF10, ARF16, and ARF17, mediated by miR160 and miR167, resulted in enhanced salinity tolerance of cotton under high salt stress. Overexpression of osa-miR319a exhibited higher tolerance to drought and salt stress by regulating the *TCP* transcription factor [28].

Increasing evidence has also revealed that miRNA-mediated gene regulation plays a significant role in heavy metal regulatory networks. In addition, high-throughput genome-wide expression profiling has greatly improved our current understanding of the key involvement of miRNA in the toxic response of plant HMs and its targets. A substantial number of heavy-metal-responsive miRNAs have been identified in the *Oryza sativa* L., *Zea mays* L., *Medicago truncatula* L., *Hordeum vulgare* L., *Vitis vinifera* L., *Brassica juncea* L., and many other plants (Table 1) [6,29,30]. In different plant species, many miRNAs have significant differential expression of different HM such as Cd, Hg, Al, As, and Cr. Through the analysis of metal-regulated miRNA target genes, it has also been identified that many miRNAs are involved in the response of plants to HM. The target genes for various processes, including metal absorption and transport, sulfate distribution and assimilation, protein folding, antioxidant systems, and plant hormone signal transduction processes [11]. We searched the Web of Science core collection database for articles related to miRNA and heavy metals such as Cu, Cd, Hg, Al, As, and Cr in the past 10 years, and a total of 568 papers were retrieved. The research fields are mainly distributed in three aspects: Life Sciences Biomedicine, Science Technology, and Physical Sciences. Then, bibliometric analysis was performed using VOS software [31], and four clusters were formed with miRNAs, stress, cadmium, and strategy as the research centers (Figure 1). These keywords are categorized by publication year to deepen the analysis. The VOSviewer software represents the year with the highest number of publications between 2016 and 2019. However, articles from 2011 to 2022 were included in this analysis. The blue circles represent topics that have been intensively studied in the past decade, such as stress response, miRNA, target gene, etc. The results of the two co-occurrence maps showed that miRNAs and stress were most closely related, suggesting that the involvement of miRNAs in stress response is the current research hotspot.

In recent years, research on miRNA and HM stress has been increasing, and, with the rapid development of advanced technologies such as next-generation sequencing (NGS), a number of known and unknown miRNAs in response to metal stress are discovered through whole genome sequencing and miRNA sequencing. However, studies on the roles of different miRNAs in HMs signal transduction in plants and their targets are still in their infancy. In this paper, we reviewed the recent research progress on miRNAs in response to plant HMs and their involvement in the regulation of target gene expression and analyzed the roles of miRNAs in HMs uptake and transport, detoxification, and plant hormone signal transduction. In addition, we explored newly discovered miRNAs and their potential targets in plants, to define the potential roles of miRNAs in plant adaptation to HM stress. This will provide clues and evidence for the regulation of gene expression under HM stress and the use of genetic engineering to improve plant tolerance to HM.

## 2. Response of miRNAs to Cu Toxicity

Cu is a trace element essential for plant growth and development. In animals, plants, and microorganisms, it is usually found in the form of Cu ions or cuprein. As an important cofactor of protein, Cu is a component of polyphenol oxidase, superoxide dismutase, laccase, cytochrome oxidase, and other enzymes, and is involved in important physiological processes such as photosynthesis, respiratory metabolism, and oxidative stress [76,77]. Cu is also a component of plastocyanin and participates in the electron transfer process of photosynthesis. However, the deficiency or excessive accumulation of Cu can cause damage to plant growth. Cu deficiency can lead to blue-green, wrinkled, distorted, or necrotic leaves, dwarf plants, slow growth, and reduced yields [78]. In contrast, excess Cu induces rapid synthesis of oxidation anions (O_2_^−^), hydroxyl radicals (OH), hydrogen peroxide (H_2_O_2_), singlet oxygen (^1^O_2_), and ROS in plant cells [79]. This reduces the strength of the cell membrane, leading to the toxic effects of Cu^2+^ infiltration into the cells. Excess Cu also inactivates chloroplast enzyme activity, accelerates chloroplast decomposition, inhibits chlorophyll synthesis, or rapidly compounds and destroys chlorophyll in plant cells. Additionally, Cu stress causes the electron-transport chain to be blocked, affecting plant photosynthesis [80]. Cu toxicity also affects the normal uptake of other mineral nutrients by plant roots. Cu stress disrupts the structure of protoplasts and affects their function, changing the permeability of cell membranes to increase, leading to the leakage of various ions from the membrane, and disrupting ionic equilibrium and a corresponding decrease in nutrient content [81]. To maintain the correct concentration of Cu^2+^ in cells, plants develop an important regulatory network in Cu uptake, distribution, and molecular responses to frequent Cu changes. The involvement of a group of miRNAs in this network appears to be particularly important, as they regulate many functionally distinct Cu proteins, including laccases, plastocyanins (PC), Cu/Zn superoxide dismutase, and polyphenol oxidase. In recent years, many studies have found that the miRNAs involved in copper stress response mainly include miR397, miR398, miR408, miR857, and miR1444, among which miR397, miR398, and miR408 are conserved in Arabidopsis and rice [32,67].

miR398 is a highly conserved miRNA in terrestrial plants, and there are three members of the miR398 family in Arabidopsis (miR398a, miR398b, miR398c) [32,67]. The sequence of miR398b and miR398c are identical, and miR398a differs from them by only one nucleotide at the 3′ end [67]. Compared with miR398b and miR398c, the promoter sequence of miR398a does not contain the GTAC sequence, which can explain the low expression of miR398a and the slow response to Cu deficiency [67]. In addition, miR398 regulates three genes through transcript cleavage and translational inhibition, including the cytosolic (*CSD1*) and plastidic (*CSD2*) genes in the Cu/Zn superoxide dismutase gene family, the copper chaperone for the superoxide dismutase (*CCS1*) gene [67]. The CSD1 and CSD2 genes encode closely related Cu/Zn superoxide dismutases, which detoxify superoxide free radicals. *CCS1* is an intracellular Cu transporter protein with the function to transport Cu ions to the *CSD* through protein-to-protein interactions, which in turn activates the *CSD*. Under high Cu stress, the expression of miR398 suppressed, resulting in an increase in *CSD1* and *CSD2* expression, which relieves the threat of ROS due to the increased Cu content [67]. In the absence of Cu, the expression of miR398 would be induced, and the transcription levels of its target genes *CSD1* and *CSD2* would decrease, while iron superoxide dismutase (FeSOD) will be up-regulated to increase the Cu availability of other important Cu proteins (such as plastocyanin) [67]. miR398 plays a key role in the regulation of Cu homeostasis by regulating the non-essential Cu protein CSD, when Cu is lacking or excessive.

Meanwhile, miR397, miR408, and miR857 are also involved in regulating the abundance of other Cu proteins in Arabidopsis, especially the effectiveness of laccase and the secreted protein plantacyanin in response to Cu [33]. As a Cu-containing oxidase, laccase can promote the synthesis of lignin from lignin monomer in plants and promote the lignification process of plants [82]. miR397 is a key regulator of Cu homeostasis in plants such as *A. thaliana*, *P. trichocarpa*, and *V. vinifera*. Under low Cu stress, the expression of miR397 was up-regulated, the synthesis of laccase or plastocyanin was inhibited, and the accumulation of Cu increased, which was beneficial to maintain the homeostasis of Cu [63,64,83]. In tomato (*Solanum lycopersicum* M.), after overexpression of miR397a, its target gene *LeLAC^miR397a^* was down-regulated, which caused a decrease in the activities of polyphenol oxidase (PPO), superoxide dismutase (SOD), and peroxidase (POD). In addition, miR397 is also related to the yield of crops. Compared with wild-type plants, transgenic rice that overexpresses miR397 has larger seeds, increased inflorescence numbers, and increased yields [59]. Studies have shown that miR408 can respond to abiotic stress and maintain Cu homeostasis in plants [69]. A study showed that the miR408 could bind to the 5`-UTR region of *LAC1*, *LAC12*, and *LAC13*, and regulated the expression of these genes to maintain Cu homeostasis in plants [63,69]. It was demonstrated that transgenic strains with simultaneous inhibition of the functions of three conserved Cu-miRNAs (miR397, miR398, and miR408) showed the reduced accumulation of Cu-miRNAs, the increased accumulation of transcripts encoding Cu proteins, and that photosynthesis and growth of transgenic plants were affected under low Cu conditions, which may be associated with defective accumulation of chloroplast plastocyanin [84]. Interestingly, miR1444 is currently only found in *P. trichocarpa* and appears to be a Populus-specific miRNA. It has also been shown to regulate a group of Cu-containing proteins: polyphenol oxidases (PPOs) [65].

The study found that the promoter regions of these several miRNAs all have Cu-response elements (CuRE) as the role sites of Cu regulation, which is composed of repeated GTAC sequences and is the only Cu-responsive element identified to date [85]. SQUAMOSA promoter-binding protein-like-7 (*SPL7*) is a key transcription factor that is sensitive to low Cu and plays a transcriptional regulatory role by binding to GTAC sequences [68]. In Cu deficiency, *SPL7* can activate the transcription of miR397, miR398, miR408, and miR857 genes, which inhibit the expression of copper-containing protein genes such as laccase and Cu-Zn superoxide dismutase, thereby regulating Cu in the case of Cu-deficiency allocation [68]. When Cu accumulates, *SPL7* becomes inactive and the expression of miRNAs in plants are inhibited, which increases the transcription level of target genes and improves the availability of Cu proteins, thus alleviating the oxidative damage caused by copper accumulation (Figure 2).

With the development of transcriptome and small RNA sequencing technology, a large number of studies will utilize this method to identify miRNAs in plants. In 2015, under the control and Cu stress conditions, the conservative and non-conservative miRNA and other short RNA were identified in *Paeonia ostia* T. [86]. A total of 102 known plant miRNAs were identified and combined with transcriptome sequencing data, while 34 new potential miRNAs were identified under the same conditions. It was also found that 12 conservative miRNA and 18 new miRNAs changed significantly under Cu stress. Jiu et al. (2019) used high-throughput sequencing to determine the miRNAs and target genes of grapevine (*V. vinifera*) plants responsive to Cu stress [87]. Among them, 100 known and 47 newly discovered miRNAs were differentially expressed under Cu stress. The target prediction of miRNA shows that miRNA may regulate transcription factors such as *AP2*, *SBP*, *NAC*, *MYB*, and *ARF* under Cu stress. They also found that miR156 targets *SPL7* and may be involved in the regulation of Cu homeostasis in grapes (*V*. *vinifera*). In addition, a total of 65 known miRNAs and 78 new miRNAs predicted to mature in mulberry were identified [30]. In total, 40 miRNAs were differentially expressed under Cu stress, of which 27 miRNAs up-regulated genes and 13 miRNAs down-regulated genes. Utilizing high-throughput sequencing technology, a large number of unknown miRNAs have been found to be involved in the Cu-stress-response process. How these miRNAs regulate the distribution of Cu in plants, enabling plants to coordinate the expression and development of Cu proteins, requires further studies to explore.

## 3. Role of miRNAs in Response to Cd Toxicity

Cd is a non-essential and highly toxic metal, which has high mobility in the soil. Cd is readily absorbed by plant root systems and transferred to aerial tissues, where Cd could accumulate in excess in edible organs such as leaves, grains, and fruit [88,89]. Therefore, it can invade the human body through the food chain, which causes serious health problems in humans, including chronic toxicity to the kidneys, bones, and lungs [90,91]. The excessive accumulation of Cd in plant cells cause several biochemical and physiological processes, mainly by disrupting enzyme-based systems [92,93]. Cd ions rapidly bind to apoplastic and symplastic proteins, including members of the heavy metal ATPase (HMA), ATP-binding cassette (ABC), natural resistance-associated macrophage protein (NRAMP), and Fe superoxide dismutase (Fe-SODs) protein families, disrupting their activity and inducing a state of oxidative stress [38,92,94]. To mitigate the damage caused by Cd, plants have evolved complex, multi-level regulatory mechanisms to cope with Cd stress. The studies on miRNAs and its related genes under Cd stress provide a basis for understanding the molecular mechanisms of plant tolerance to HM tolerance.

In plants, metal transporters are key components responsible for metal uptake, translocation, and homeostasis. Cd uptake and transport in plants are mediated by NRAMPs and ABC metal transporters [49]. NRAMPs are a highly conserved family of membrane proteins involved in the transport of metal ions in most organisms. The ABC carrier is a transporter located on the organelle membrane, which plays a role in the form of heavy metal chelate in the process of absorbing Cd in the vacuole. These are thought to be regulated by miRNAs (Figure 3). In addition, genes encoding ABC transporter protein and NRAMP1b metal-transporter protein were shown to be targeted by miR159 and miR167, respectively, and to respond to Cd stress [46,95]. In rice, a total of 12 Cd-stress-related miRNAs were identified and processed, among which miR268 was significantly up-regulated under Cd stress [51]. It was later found that miR268 negatively regulated the expression of NRAMP3 under Cd stress and that overexpression of miR268 inhibited rice seedling growth under Cd-stress treatment [52]. The overexpression of miR192 decreased the expression of *ABC* gene, and miR192 had negative effects on the seed germination of rice under Cd stress [96]. The homeodomain containing protein 4 (*OsHB4*) gene encodes the HD-Zip protein and participates in the process of stress response to adversity, which was up-regulated by Cd treatment, though the expression of miR166 was decreased. The overexpression of miR166 down-regulated *OsHB4* expression, reducing Cd translocation from roots to shoots and Cd accumulation in seeds and decreasing Cd-induced oxidative stress [45]. It can be seen that miRNAs can directly regulate the expression of Cd transport genes or indirectly affect Cd transport to reduce Cd transport and accumulation, thereby alleviating the damage caused by Cd stress.

Cd stress leads to the excessive accumulation of ROS in plants, which damage lipids and protein nucleic acids and influence the growth and development of plants, while miRNAs play a key role as post-transcriptional regulators of genes involved in antioxidant response pathways [97]. miRNA eliminates ROS by targeting and regulating the expression of related genes, such as superoxide dismutase (SOD), catalase (CAT), ascorbate peroxidase (APX), non-enzymatic components, ascorbic acid (AsA) and GSH [34,52]. After treating wheat seedlings with 150 μM Cd solution, the expression of miR398 in wheat leaves decreased, while the expression of its target gene *CSD* increased. The decrease in miR398 levels leads to the accumulation of *CSD* mRNA, which plays an important role in catalyzing the mutation of superoxide radicals into H_2_O_2_ [98]. A study discovered that overexpressing miR156 (miR156OE) accumulated significantly less Cd in the Arabidopsis shoot, increased the activities of antioxidative enzymes, and reduced the level of endogenous ROS, which further improved the tolerance to Cd stress [34]. Furthermore, high-affinity ligands are used to chelate Cd and reduce its biological toxicity. Sulfur (S) is an essential macronutrient for plants, and sulfate (SO_4_^2−^) is the main form of inorganic sulfur nutrient in the soil, which plays a role in plant responses to Cd exposure [11]. Sulfate is assimilated in plant cells with the help of a number of key enzymes to produce the sulfur-containing metabolites cysteine and GSH, as well as PCs and other sulfur-containing compounds, which play a key role in plant chelation and antioxidant defense against Cd toxicity [99,100]. Research indicates that miRNAs alleviate the Cd stress causing oxidative damage by targeting the related genes that regulate this process. Its absorption and assimilation are particularly important for the metal tolerance of plants. Studies have shown that sulfate transporter 2; 1 (SULTR2; 1) and ATP sulfonylase 1, 3, 4 (APS 1, 3, 4) are sulfate-assimilation-related genes that can be targeted and regulated by miR395, which proves that miR395 is involved in the process of sulfur assimilation [60,101,102,103]. In *B. napus*, overexpressing miRNA395 can strengthen Cd tolerance by regulating the increase in the pure content of GSH and non-protein sulfur [60]. Consistent with this, when the high-Cd-accumulation variety *B. parachinensis* was treated with Cd stress, the enhancement of miR395 promoted the accumulation of Cd, but these plants showed less metal toxicity [96]. Zhou et al. demonstrated that this is probably due to the sulfur assimilation regulated by miR395 involved in the Cd accumulation and detoxification of high-Cd-accumulating varieties [96]. Moreover, the induction of miR395 expression is controlled by sulfur-restricted 1 (SLIM1), SLM1 is a transcription factor for assimilating sulfur metabolism in higher plants and regulates the major pathway of sulfate uptake and metabolism under the -S environment in Arabidopsis roots. The miR395 acts downstream of the SLIM1 transcription factor and upstream of *APS* and *SULTR2;1*, to control the distribution and assimilation of sulfate, thereby improving plant tolerance to Cd stress [1,61].

Cd-stress signals are closely related to the endogenous levels of plant-growth regulators (PGRs). Numerous studies have shown the potential role of PGRs such as auxin, cytokinin (CK), and jasmonic acid (JA) in the regulation of Cd tolerance in plants [56,104,105,106,107,108]. Some recent approaches showed that exogenous addition of auxin or stimulation of endogenous levels can prevent growth inhibition and improve heavy metal tolerance, and miRNAs appear to be involved in the regulation of auxin homeostasis. High-throughput RNA sequencing studies in *Z. mays*, *Sedum alfredii H.*, and *O. sativa* have identified a large number of Cd-responsive miRNAs, some of which are associated with auxin signaling [56,60,61,100,101,102,103,104,105,106,107,108]. Research showed that miR393 functions in plant Cd-stress tolerance, which targets transport inhibitor response 1 (TIR1) in rice [100]. TIR1 encodes an E3 ubiquitin ligase containing a F-box subunit, acting as an auxin receptor by interacting with Aux/IAA proteins [109]. Therefore, the induction of miR393 exacerbates TIR1 levels during Cd stress. This ultimately leads to the down-regulation of auxin signaling pathway and may result in the reduction in E3 ubiquitin ligase targeting protein hydrolysis. It indicates a potential crossover between auxin signaling and Cd stress signaling [110].

It has also been noted that CKs in plants are activated under HM stress and are able to reverse HM-induced toxicity. Application of exogenous CKs counteracted the inhibitory effects of Cd on growth and photosynthesis and enhanced photosynthetic capacity and primary metabolite levels through the CK-mediated induction of plant metabolism [75]. In addition, miR1535b, a Cd-response miRNA in soybean, was predicted to be cleaved by *Glyma07g38620.1*, which encodes isopentyl transferase (IPT). IPT is an important rate-limiting enzyme for the synthesis of CTKs, which catalyzes the rate-limiting first step in de novo CK biosynthesis and promotes the formation of isopentenyladenosine-59-monophosphate (iPa). Overexpression of iPT in roots and leaves demonstrated the improvement of stress tolerance ability in *Arachis hypogaea* L. The up-regulation of *Glyma07g38620.1* under Cd stress showed its positive correlation with CK production [111]. At present, there are few studies on the regulation of plant response to Cd by miRNAs involved in the CK signaling pathway, so the related signaling pathways need to be further explored. Furthermore, recent reports found that JA is related to the response of plants to Cd stress, the expression of endogenous JA synthesis gene was rapidly induced after Cd treatment [112,113,114]. Tuli et al. (2010) have reported that miR168b in rice targets lipoxygenase 1 (*LOX1*), which encodes an important enzyme in JA biosynthesis. Interestingly, miR168 was identified as a Cd-responsive miRNA [48,115]. So, the miR168b may regulate endogenous JA synthesis by targeting the *LOX1* gene, regulating the plant response to Cd stress. We all recognize the importance of plant growth hormones (such as IAA, CK, and JA) in alleviating Cd stress, so it is particularly important to further investigate how miRNAs regulate plant tolerance to Cd stress by regulating genes related to the phytohormone signaling pathway.

## 4. Involvement of miRNA in Response to Hg Toxicity

Hg as one of the toxic heavy metal elements, and it is the only metal element that exists in liquid form under normal temperature and pressure. It mainly exists in three forms: metallic Hg, inorganic Hg, and organic Hg (methyl Hg, etc.) in nature [116,117]. Environmental media such as the atmosphere, water, and soil all contain Hg. Among them, ionic Hg (Hg^2+^) can be absorbed by plants due to its solubility and mobility [118]. A small amount of absorption will cause greater damage. Since Hg can directly bind to biological macromolecules (such as the sulfhydryl groups of proteins), it disrupts the function of biological macromolecules and disrupts the normal structure of plasma membranes [119]. Hg stress will stimulate the increase in ROS content in plants, leading to the occurrence of oxidative stress, which plays a major role in causing plant damage [120]. In addition, Hg stress can also inhibit the germination of seeds, affect photosynthesis, and even cause plant death [121]. Therefore, in-depth study of the regulatory mechanism of the post-transcriptional level is of great significance for reducing the toxicity and accumulation of Hg in plants.

More and more studies have reported that miRNAs and their target genes are involved in the regulation of plant Hg tolerance. Using biological information prediction and RT-PCR technology, a set of miRNAs in response to Hg stress was identified from *M*. *truncatula* for the first time [50]. After Hg stress, the expressions of miR171, miR319, and miR393 were up-regulated, and the expressions of miR166 and miR398 were down-regulated, while the expressions of miR160 and miR395 were not affected [50]. Zhou et al. described 210 known miRNAs and 54 candidate miRNAs in *M*. *truncatula* seedlings [122]. The results showed that the expression of 15 conserved miRNA families was up-regulated, and 4 conserved miRNA families were down-regulated. Most miRNAs target genes coding for tolerance proteins or enzymes. For example, miR2681 targets several transcripts encoding TIR-NBS-LRR disease-resistant proteins. A salt-tolerant protein (TC114805) is the target of miR2708. It is worth noting that miR2687 targets the gene encoding xyloglucan endotransglucosylase/hydrolase (XTH), which is considered to be a cell-wall-modifying enzyme, which participates in cell wall development and gives plants resistance to abiotic stress. A study also showed that OsPDIL1 is an important protein folding catalyst in rice and that miR5144 (osa-miR5144-3p) mediates the formation of protein disulfide bonds by targeting *OsPDIL1;1* mRNA. Compared with wild-type, the down-regulating osamiR5144-3p, or overexpressing *OsPDIL1;1*, transgenic rice line increased total protein–disulfide bond content and improved tolerance to Hg stress [123]. Therefore, the *OsPDIL1;1* over-expressed rice line and the osa-miR5144-3p down-regulated rice line were resistant to abiotic stress and accumulated lower Hg content, indicating that they may be useful traits for rice molecular breeding. Therefore, further identification and functional verification of miRNAs and targeted functional verification have prompted us to further explore the mechanism of plant Hg tolerance.

## 5. Response of miRNAs to As Toxicity

As is a widely distributed metalloid element with strong toxicity. It is one of the most common environmental chemical pollutants with the most serious health hazards for residents. As exists in organic and inorganic forms, with inorganic As occurring in nature in two main forms, arsenate As (V) and arsenite As (III). According to the changes of oxidation-reduction potential and pH, these two inorganic As species can easily transform into each other [124]. Under reducing conditions, As (III) is the main form, while As (V) is the stable form in an oxygen-containing environment [124]. Both the inorganic forms of As are highly toxic, one of which (As (III)) interacts with and binds to the sulfhydryl group of the protein, thereby inhibiting its function. On the other hand, As (V) interferes with phosphate metabolism by inhibiting phosphorylation and ATP synthesis [125,126]. As inhibits the germination of plant seeds, affects root formation and shoot growth, and also affects plant photosynthesis, respiration, transpiration, water metabolism, energy metabolism, carbon and nitrogen metabolism, and other processes [127]. The adverse effects of As stress on plants are mainly due to the presence of As that causes harmful ROS to cells. The decreases in the activity of CAT and SOD, the increases in the content of malondialdehyde (MDA), and the damage to the membrane system in the body all affect the absorption and interference of other elements by plants [66]. The transportation of other substances by plants, which in turn affects the physiological and metabolic functions, then affects the normal growth of plants [70]. As has many effects on plant growth and development. Plant response to As stress is a comprehensive response of various physiological processes. The study of miRNAs related to As stress provides a feasible solution for clarifying the molecular mechanism of As accumulation in plants and solving the problem of As pollution.

Liu and Zhang (2012) found that 54 and 13 miRNAs belonging to 19 and 7 miRNA families in rice were significantly down-regulated and up-regulated after As stress, respectively [43]. The expression of miR408, miR528, and miR397b were significantly up-regulated under As stress, while the expression of miR1318 and miR390 was repressed. The predicted target genes of miRNAs showed that the functions of target genes of As stress-related miRNAs were mostly focused on cellular processes and metabolic processes. Sharma et al. (2015) compared the response and accumulation of two rice varieties to As (III) and As (V), and identified differentially expressed miRNAs in response to As (III) and As (V) [128]. The results indicated that miR396, miR399, miR408, miR528, miR1861, miR2102, and miR2907 family members were significantly up-regulated in response to both As (III) and As (V) stress, while miR164, miR171, miR395, miR529, miR820, miR1432, and miR1846 family member were inhibited. Sequence identification indicated that metal-responsive cis-acting elements as well as other hormone-related motifs were present in the promoters of all these miRNAs. In 2012, Yu et al. studied the As (III)-responsive transcriptome (including mRNA and miRNAs) of rice roots and shoots using Illumina high-throughput sequencing (HTS) technology. Overall, 14 of the 36 As (III)-responsive miRNA are thought to be involved in the regulation of As (III)-responsive genes, and these genes responding to As (III) stress are mainly functionally enriched in heavy metal transport, JA signaling, or lipid metabolism [129]. Based on the sRNA HTS data reported above, Tang et al. (2019) conducted a comprehensive search for As-responsive sRNAs in rice roots and shoots and identified 37 As-responsive sRNAs sharing the same sequence with 59 miRNAs from 26 miRNA families [47]. Additionally, 14 of the 26 miRNA families were newly identified As-responsive families, which may be due to the result of miRBase updates.

In 2009, Srivastava et al. discovered that the increase in JA levels stimulates the sulfate assimilation pathway and activates the detoxification process to achieve effective As complexation [44]. Srivastava et al. performed a microarray analysis of the miRNAs in mustard and found significant alterations in the expression of 69 miRNAs from 18 miRNA families [39]. In addition, they found that the supply of exogenous JA and IAA could improve plant growth under As stress, altering the expression of miR167, miR319, and miR854, which indicated that hormones and miRNAs have crosstalk in regulating the plant response to As stress. The study has also reported that miR838 targets a gene encoding a lipase that plays an important role in oxylipin and JA biosynthesis. miRNAs respond to As stress mainly by regulating the expression of metal transporters. OPT1 is a hypothetical target of miR159 under As stress [39]. Other OPTs have been confirmed to be involved in the transport of Fe, Zn, Cu, Ni, Mn, and Cd, indicating that they may play a role in As toxicity. Moreover, the study has showed that miR528 is closely related to As tolerance [29]. It was reported that an over-expressed miR528 transgenic rice line was more sensitive to As, exhibiting more pronounced oxidative stress and significant changes in proline content of roots and leaves. The prediction results showed that miR528 targets the transcription products of two one plastocyanin-like protein gene (*LOC_Os07g38290.1*), two multi-copper oxidase genes (*LOC_Os01g03640.1*, *LOC_Os01g03620.1*), two laccase-encoding genes (*LOC_Os01g62600.1*, *LOC_Os01g44330.1*), and two galactosyltransferase family protein-coding genes (*LOC_Os12g41956.1* and *LOC_Os07g09690.1*) [29]. The direct relationship between these genes and As stress needs to be further verified.

Under As stress, the expression analysis of OsamiR156j successfully demonstrated its importance in various developmental stages and tissues of rice. The presence of target function and cis-regulatory elements/motifs in response to oxidative stress also confirmed that miR156j is involved in the regulation of As stress [130]. It can be seen that there are a large number of studies on the regulatory role of miRNAs in the process of plant As poisoning, especially miRNAs that play an important role in the regulation of JA biosynthesis, which may provide a direction for our follow-up research.

## 6. miRNA in Plants during Cr stress

With the development of electroplating, mining, smelting, pharmaceuticals, and other industries, about 10 million tons of chromium-containing wastewater are discharged into the environment every year, meaning Cr pollution has become a serious heavy metal pollution [130]. Cr is the seventh-most-abundant element in the Earth’s crust and is usually found in soil in the stable forms of trivalent Cr (III) and hexavalent Cr (VI) [54]. Cr (Ⅵ) is the most toxic valence form of Cr, which is easily soluble in water. The compound formed is less soluble, more stable, easy to migrate in the soil environment, and relatively more toxic [53,131]. However, Cr (Ⅲ) has low activity, poor mobility, and relatively low toxicity and is easy to precipitate and be adsorbed by soil colloids [131,132]. For plants, the low concentrations of Cr could promote plant growth and increase yield, but Cr is a non-essential element for plant growth. Excessive Cr will inhibit seed germination, nutrient balance, and enzyme activity, reduce root growth and biomass, and induce plant leaf chlorosis and oxidative stress, even causing plant death in severe cases [66]. Therefore, analyzing the accumulation of Cr and its regulation mechanism at the post-transcriptional level is of great significance for reducing the detrimental effects of Cr in plant tissues. The identification of Cr-stress-related miRNA and related genes is helpful to understand the molecular genetic mechanism of plants in response to Cr stress.

Liu et al. (2015) investigated the changes in the expression of miRNAs under Cr stress in radish (*Raphanus sativus* L.) and found that 54 known miRNAs and 16 novel miRNAs were differentially expressed under Cr stress. The expression of 37 miRNAs (28 known miRNAs and 9 novel miRNAs) were up-regulated, and the expression of 33 miRNAs (26 known and 7 novel miRNAs) were down-regulated [131]. The target genes of Cr-responsive miRNAs encode different families of transcription factors, including *SPLs*, *MYBs*, *ERFs*, and *bZIPs*, which may regulate the corresponding HM-related transcriptional processes in plants. It is worth noting that some key reaction enzymes or proteins, including HMA, YSL1, and ABC transporters, are involved in the uptake and homeostasis of Cr [132,133,134]. In rice, 512 and 568 known miRNAs were identified from Cr-treated and untreated samples, respectively [35]. The results revealed that 13 conserved miRNAs (miR156, miR159, miR160, miR166, miR169, miR171, miR396, miR397, miR408 miR444, miR1883, miR2877, and miR5072) exhibited preferential up-regulation or down-regulation. Prediction of target genes of differentially expressed miRNA and their functional annotation indicated that miRNAs play critical role in Cr defense and detoxification through ATP-bound cassette transporters (ABC transporters), transcription factors, heat shock proteins, growth hormone responses and metal ion transport. Additionally, 53 known miRNAs and 29 unknown miRNAs were identified in tobacco, including miR156, miR159, miR166, miR167, miR171, miR396, and miR399 [36]. miRNA sequencing was performed on *M. sinensis* under Cr treatment, and a total of 104 conserved miRNAs and 158 non-conserved miRNAs were identified. In total, 45 miRNAs were differentially expressed in roots, and 13 miRNAs were differentially expressed in leaves. Based on candidate-gene annotation and GO and KEGG function analysis, miR167a, novel_miR15, novel_miR22, and their targets may be involved in the transport and chelation of Cr. Furthermore, miR156a, miR164, miR396d, and novel_miR155 are involved in the physiological and biochemical metabolism and the detoxification of Cr in plants [37]. These results indicate that miRNAs may respond to Cr stress by regulating metal transporters, transcription factors, hormone signal regulation, and other pathways, providing new insights for exploring the response mechanism of plants to Cr stress and helping to develop new strategies for plants to combat heavy metal stress.

## 7. miRNA Involved during Al Stress in Plants

Al toxicity is the main factor restricting crop yields, especially on acidic soils that account for 30–40% of the world’s arable land [2]. Soils with a pH value of 5.5 and below are susceptible to the effects of Al toxicity, because the combined Al will dissociate into free Al ions (mainly trivalent Al ions, Al^3+^), the ionic form of Al is toxic to plants [135]. Al toxicity can interfere with DNA synthesis and mitosis, destroy the structure of the Golgi apparatus and inhibit mitochondrial function, affecting the cell cycle and intracellular activities [136]. The most typical and visible effect of Al on plants is to inhibit root elongation. The root-elongation process includes cell elongation and cell division. Al toxicity firstly affects the cell-elongation process, and the cell-division activity will also be inhibited in the later period [136]. The root system is destroyed, and the absorption of water and nutrients is restricted. Long-term Al toxicity makes the plant stems shorter and the leaves short and yellow, and even the seedlings die, which ultimately leads to reduced crop yields [136,137]. In addition, Al toxicity also affects several other physiological processes. High concentrations of Al have damaging effects on different physiological pathways such as callose deposition, cytoplasmic Ca^2+^ level differences, and oxidative stress [137].

Chen et al. (2012) identified 23 Al-responsive miRNAs in 7-day-old *M. truncatula* seedlings after 10 uM AlCl_3_ treatment [40]. Based on expression-pattern analysis, they further classified 18 miRNAs as fast-response miRNAs and 4 miRNAs as sustained-response miRNAs, miRNA390 is the only late-response miRNA. Among the down-regulated miRNAs, five miRNAs (miR159, miR160, miR319, miR390, and miR396) have been reported to have known functions. Subsequently, Zeng et al. (2012) identified 30 Al-responsive miRNAs in wild soybean root systems, of which 10 conserved miRNAs belonged to 7 conserved miRNA families, 13 non-conserved miRNAs, and 7 novel miRNAs [55]. However, the expression of miR396 and miR390 was up-regulated in *M. truncatula* after Al stress, which is different from the results of Chen et al. (2012). The difference may be due to the discrepancy in the genomic and tolerance mechanisms of *M. truncatula* and *G. max*. In 2019, Silva et al. used root systems of Al-tolerant and sensitive sugarcane varieties to generate four miRNAs libraries under Al stress [41]. In total, 394 differentially expressed miRNAs were identified in the two varieties, of which 104 were expressed specifically in the Al-tolerant variety, 116 in the Al-sensitive variety, and 87 were commonly expressed between the two varieties. In the Al-tolerant variety, the expression of miR159, miR160, miR393, miR398, and miR164 was all down-regulated. The down-regulation of miR398 attenuated ROS-induced oxidative stress, while the repression of miR159, miR160, miR393, miR121, and miR164 regulated signal transduction, root development, and lateral root formation.

Studies have found that auxin-related genes are regulated by most miRNAs under Al stress. miR160 regulates the auxin signal through auxin-response factors (*ARF6* and *ARF10*) to control root and cap development, and miR160 can regulate the damage of root growth under aluminum stress by targeting the expression of *ARF* [40,42]. miR390 can negatively regulate the growth of lateral roots by generating tasiRNA from TAS3 transcripts to indirectly degrade *ARF* [138]. Another regulator of auxin-related genes is miR393, which targets the F-box auxin receptor TIR1/AFB. TIR1 is a positive regulator of auxin signaling. However, in the presence of Cd, Hg, and Al, the increase in miR393 expression will cause a decrease in *TIR1* levels and inhibit root growth and root structure changes [57,139]. Bai et al. (2017) observed that miR393 induced under Al stress leads to the changes in root morphology through altered auxin sensitivity, and the reduction in ROS-induced cell death was observed following miR393 overexpression [58]. In addition, the down-regulation of miR393b in rice may be related to the increase in proteasome-mediated protein processing under Al toxicity. Moreover, the simultaneous up-regulation of miR160e against *ARF* may be negatively related to miR393b in the auxin signaling pathway under Al stress [110].

Previous studies have shown that miR528 is involved in the regulation of cell division in *Arabidopsis* shoot branching and the early growth stage in rice seeds, by targeting the F-box leucine-rich repeat protein (F-box/LRR) repeat *MAX2* gene and L-resistant oxidase, respectively [71,72]. In maize, zma-miR528 targets the MATE family member GRMZM2G148937, which is a factor in plant Al tolerance, and it was found that the transcripts of target genes are up-regulated in response to Al stress [73,140]. The regulation of cell-wall structure in response to Al stress is also related to the targeting of rockweed glycosyltransferase in rice roots mediated by miR808 [74]. miR395 is regulated by Al toxicity and sulfate deprivation, inhibiting its low-affinity SULTR2;1 target. In addition, sulfate deprivation induces high affinity SULTR2;1, which improves the absorption of sulfate from low-sulfate soil solutions [62]. Examples of cross-tolerance have also been observed in the case of conserved miRNAs under different metal stresses and abiotic stresses. miR156 was down-regulated under Cd stress and also showed a down-regulation pattern under Al stress [40,48]. miR164 induces the *NAC1* TF gene under drought stress, leading to the down-regulation of growth hormone signaling, which shows a down-regulation pattern under Al stress. In the case of conserved miRNAs under different metal and biological stresses, examples of cross-tolerance have also been observed. miR156 was down-regulated under Cd stress [48] and also showed a down-regulation pattern under Al stress [40]. miR164 induced and targeted *NAC1* TF under drought stress, which resulted in the down-regulation of auxin signaling, showing that it was down-regulated under Al stress [55,141]. These results indicate that miRNAs intercepted different signals and transport steps by regulating different developmental stages of plants to improve the tolerance of plants to Al stress.

## 8. Conclusions and Perspectives

Data shows that nearly one-fifth of Chinese cultivated land area (about 2 million hm^2^) is polluted by HM such as Cd, Hg, Pb, and metalloid As [58]. HM pollution not only seriously affects the yield and quality of crops but also can cause harm to human health through accumulation in the food chain. For example, growth and yield were significantly reduced in both maize and rice when exposed to Cd stress, inhibiting seed germination, retarding root and seedling growth, and suppressing photosynthesis and respiration [142,143]. In soybeans treated with arsenate (As V) and arsenite (As III), the root absorption rate was reduced, and the relative water content and stomatal conductance of leaves were significantly reduced [144]. In addition, high concentrations of metals Cd, Fe, and Zn inhibited wheat and legume seed germination, root length, and shoot length, with cadmium being the most toxic [145]. Therefore, exploring the regulatory network mechanism of HM uptake, transport, chelation, and detoxification in plants is beneficial for plants to cope with HM stress. Over the past decade, through high-throughput sequencing and large-scale data analysis of miRNA chips, a large number of HM-sensitive miRNAs have been discovered in rice, maize, barley, alfalfa, grape, poplar, and many other plants. In different plant species, many miRNAs are significantly differentially expressed for different HMs such as Cd, Hg, Cu, As, and Pb. A large number of studies have demonstrated that miRNAs are key components of the transcriptional regulatory network in plant heavy metal stress response.

miRNAs are involved in HM uptake and transport, sulfate partitioning and assimilation, protein folding, antioxidant systems, and phytohormone signaling, by regulating the expression of target genes to facilitate plant responses to HM stress. miRNAs participate in the process of heavy metal absorption, transport, and balance, by regulating *NRAMP1*, *NRAMP3*, *OPT1*, *HB4*, and other genes [44,46,51,96]. miR395, miR398, miR164, and miR528 are involved in the process of sulfur absorption and assimilation, and promote the production of antioxidant enzymes such as GSH, PC, and SOD, thereby reducing the oxidative damage caused by HM stress [67,98,99,103]. miRNAs may maintain the balance of plant hormone levels by regulating the expression of genes related to the synthesis pathway of IAA, JA, and CK, thus effectively resisting heavy metal stress [22,44,56,95,109,111,146,147,148].

Current studies have confirmed that miRNAs are involved in regulating HM-response processes, by further understanding the mechanisms of HM regulatory networks, which will pave the way for improving the function of crop tolerance to HMs. A great number of miRNAs have been identified in a variety of plants that are involved in a variety of HM-stress responses. However, most of these miRNAs have not been studied in depth, so their specific functions are still unknown. Based on previous research results, we can observe that many miRNA target genes encode transcription factors or enzymes, which also implies that miRNAs are key players in the regulatory network of HM-stress response. Additionally, there are many target genes of miRNAs that can only be predicted by bioinformatics software and have not been experimentally verified, so further experimental verification is required to link them to the intermediate pathway of HMs tolerance. Validation of miRNAs and associated targets will provide ample evidence for their role in HM tolerance, using technologies such as reverse genetics and miRNA and degradomes sequencing. Moreover, the CRISPR-Cas9 system and short tandem target simulation (STTM) technology can be used to cause miRNA function loss or overexpressed target genes, to further explore how miRNA regulates target genes and participates in the HM-stress regulatory network. It is worth noting that not all miRNA targets are conserved in different plants, and miRNA targets need to be validated in different plant species. Since a miRNA may have multiple target genes, and a gene may be regulated by multiple miRNAs, this many-to-many relationship increases the difficulty of analyzing the specific regulatory relationship between miRNAs and target genes. Furthermore, understanding the changes in the DNA methylation profiles of plant genomes under HM stress and their interactions with miRNAs will be an interesting research area in the future. How miRNAs themselves are regulated in response to HM remains a mystery and needs to be solved in the future. However, it is undeniable that miRNAs provide feasible ideas and methods for the study of the molecular mechanism of plant resistance to heavy metal stress as well as ideas for improving the stress resistance of crops through genetic manipulation of small molecules.

## Figures and Tables

**Figure 1 ijms-23-08642-f001:**
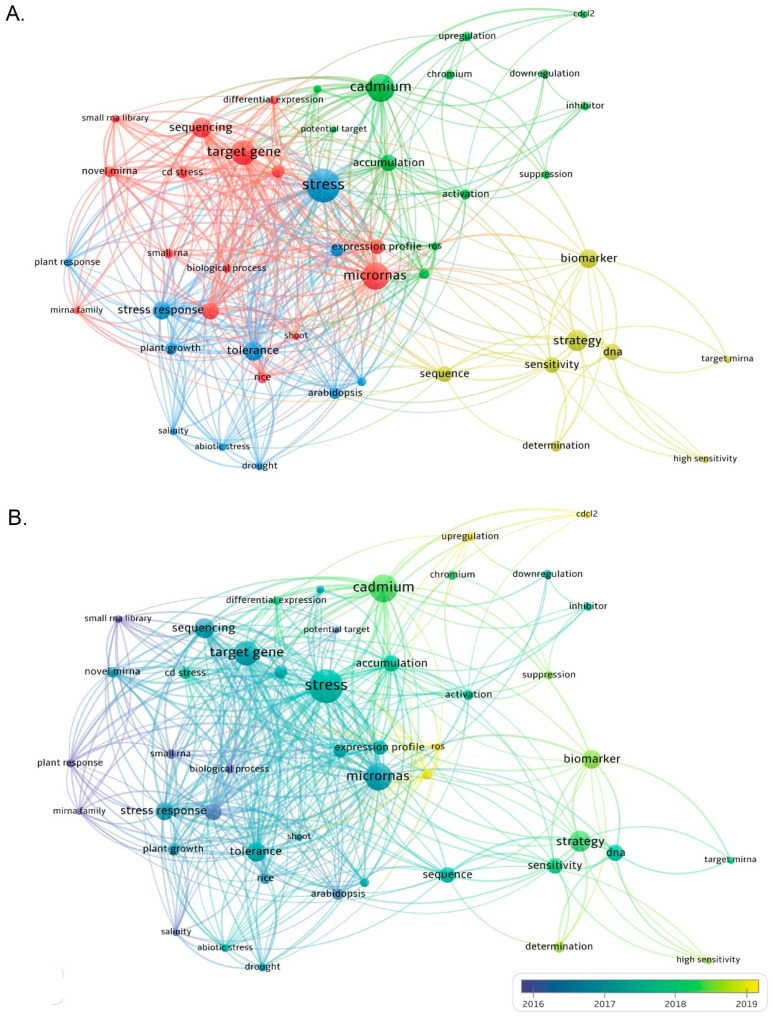
Keyword co-occurrence graph analysis using VOSviewer software. (**A**) A total of 568 relevant articles were searched from academic papers over the past 10 years using the keywords miRNA, Cu, Cd, Hg, Al, As and Cr, four research centers focusing on miRNA, stress, cadmium, and strategy were formed. (**B**) Different colors correspond to the year in which the keywords appeared on average, and keywords with blue color presented earlier than those with yellow.

**Figure 2 ijms-23-08642-f002:**
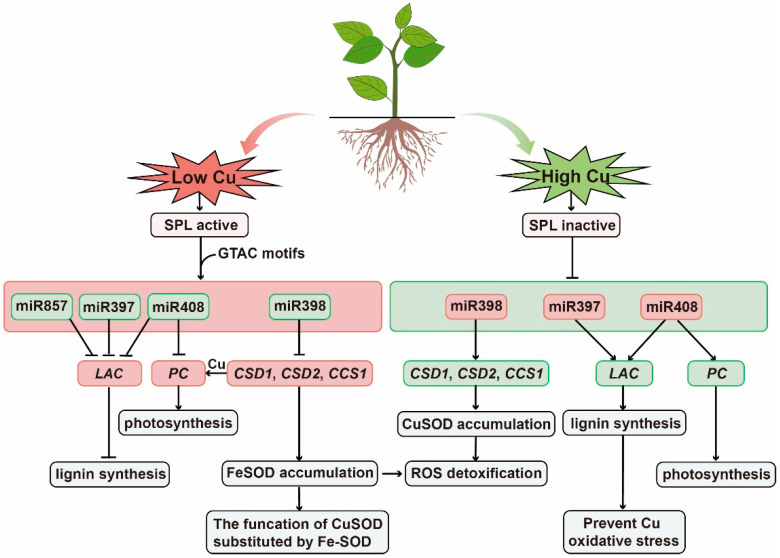
Schematic representation of the mode of action of miRNAs in response to Cu stress. Green boxes indicate relevant miRNAs and pathways involved in low Cu stress. *SPL7* transcription factors are active and regulate miR397, miR398, miR408, and miR857, which regulate genes encoding Cu-containing proteins, including Cu/Zn superoxide dismutase (*CSD*), *CCS1*, laccase (*LAC*), and phycocyanin (*PLC*), thereby conserving Cu as essential Cu protein (such as plastocyanin). Red boxes indicate that under high Cu stress, *SPL7* is inactivated, miR398 is down-regulated, and the expression of target genes *CSD1*, *CSD2*, and *CCS1* is up-regulated, mitigating the threat of ROS from increased Cu content and increased Cu protein accumulation.

**Figure 3 ijms-23-08642-f003:**
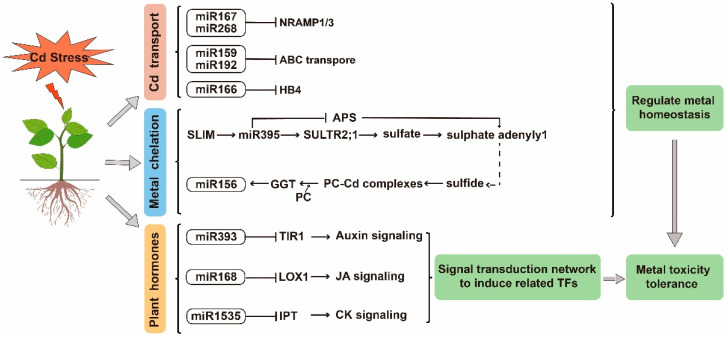
The regulatory network of miRNA target genes is involved in the response of plants to Cd stress. The blue area indicates the pathway by which miRNA participates in metal chelation. The green region represents the process by which miRNA participates in metal transport. The orange area indicates that miRNA participates in the regulation of related plant hormone signaling pathways, including auxin, jasmonic acid, and cytokinin.

**Table 1 ijms-23-08642-t001:** List of miRNAs along with their target function during heavy metal toxicity tolerance.

miRNA	Plant Species	Stress	Target Genes	Target Function	Reference
miR156	*A. thaliana, V. Vinifera, O. sativa, Nicotiana tabacum* L., *Miscanthus sinensis* A.	Cu, Cd, Cr, Al	*SPL7*	Regulate Cu homeostasis, decrease endogenous ROS	[32,33,34,35,36,37]
miR159	*Brassica napus* L., *O. sativa, N. tabacum, M. truncatula, Saccharum spp*	Cd, As, Cr, Al	ABC transporter protein, *OPT1*	Regulate metal transporters	[35,36,38,39,40,41]
miR160	*A. thaliana, O. sativa, Saccharum spp*	Cr, Al	*ARF*	Regulate auxin signal	[41,42]
miR164	*A. thaliana, O. sativa, M. sinensis,* *Saccharum spp*	As, Cr, Al, Hg	NAC, CUP trancription factors	Signaling pathway, root development, response to oxidative stress	[41,43,44]
miR166	*O. sativa, N. tabacum*	Cd, Cr	Homeodomain containing protein 4	Reduced Cd translocation from roots to shoots and Cd accumulation in seeds, decrease Cd-induced oxidative stress	[35,36,45]
miR167	*O. sativa, B. juncea, N. tabacum*	Cd, As	NRAMP1b metal transporter protein	Metal uptake and translocation	[36,46,47]
miR168	*O. sativa,*	Cd	*LOX1*	Promote JA synthesis	[48]
miR169	*M. truncatula, O. satva, N. tabaccum, B. juncea*	Cr, Cd, Al, As	CCAAT-binding (TF)	?	[35,38,44,49]
miR171	*M. truncatula, B. Juncea, O. sativa,* *N. tabacum*	Hg, As, Cr, Cd	SCL (TF)	Shoot branching, signaling pathway	[2,35,36,50]
miR192	*O. sativa*	Cd	*ABC* gene	Seed germination of rice under Cd stress	[51]
miR268	*O. sativa*	Cd	*NRAMP3*	Inhibited rice seedling growth under Cd-stress treatment	[52]
miR319	*M. truncatula, B. juncea, M. truncatula*	Hg, As, Al	TCP (TF), cyclin	Leaf morphogenesis, cell differentiation, embryonic development, cell division	[47,50,53]
miR390	*O. sativa, M. truncatula, Glycine max* L.	As, Al	tasi-RNA	Plant development	[40,54,55]
miR393	*A. thaliana, O. sativa, M. truncatula, Saccharum spp, H. vulgare*	Cd, Hg, Al	TIR1/AFBs (F -box auxin receptors) and bHLH (TF)	Regulate auxin signaling	[41,50,56,57,58]
miR395	*A. thaliana, B. napus, Brassica parachinensis* L.	Cd, As, Al	*SLIM1*, *SULTR2;1*, *APS*	Sulfur assimilation, response to cadmium ion, sulfate transport	[43,59,60,61,62]
miR396	*O. sativa, N. tabacum, M. sinensis, M. truncatula, G. max*	As, Cr, Al	GRF (TF)	Translation, leaf development	[36,37,40,43,55]
miR397	*A. thaliana, O. sativa, Populus trichocarpa* T.	Cu, As, Cr	Laccase	Regulate inter-tissue lignification and secondary cell wall thickness, activities of PPO, SOD, and POD	[35,59,63,64,65,66]
miR398	*A. thaliana, O. sativa, P. trichocarpa*	Cu, Cd	*CSD1*, *CSD2*, *CCS1* and *COX5b-1*	SODs relive oxidative stress	[34,65,67,68]
miR399	*O. sativa, N. tabacum*	As	?	?	[36,43]
miR408	*A. thaliana, O. sativa, P. trichocarpa, O. sava*	Cu, Cr, As	Laccase, plantacyanin transcripts, *SPL7*, *HY5*	Regulate plastocyanin (PC) content	[63] [35,43,65,69,70]
miR444	*O. sativa*	Cr	?	?	[35]
miR528	*O. sativa, A. thaliana, Z. mays*	As, Al, Cd	*MAX2* gene and l-resistant blo, LACod acid oxidase, *MATE*, *LAC*	Signal transduction, regulation of cell cycle, plant development, ascorbate metabolism, miRNA processing, control of cellular-free auxin levels	[43,54,70,71,72,73]
miR529	*O. sativa,* *M. truncatula*	As, Cd	Apetala2-like (TF), squamosa promoter binding protein-like (TF)	Signaling pathway, plant development	[43,50]
miR808	*O. sativa,*	Al	Rockweed glycosyltransferase	?	[74]
miR854	*B. juncea*	As	Putative serine acetyl transferase (SAT)	Synthesis of *O*-acetylserine	[47]
miR857	*A. thaliana,*	Cu	Laccase (LAC7)	?	[33]
miR1318	*O. sativa*	As	Calcium-binding proteins or Ca^2+^ ATPase	Signaling	[70]
miR1444	*P. trichocarpa*	Cu	Cu-containing proteins, polyphenol oxidases (PPOs)	?	[65]
miR1535b	*G. max*	Cd	*Glyma07g38620.1*	Responsible for initial step of isopentenyl transferase	[75]

## Data Availability

Not applicable.

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
