# Peer review of "microRNAs: Key Players in Plant Response to Metal Toxicity"

_ijms, 2022, doi:10.3390/ijms23158642_

Round 1

Reviewer 1 Report

The manuscript is well written, providing a solid portion of extracted and initially digested  knowledge. I found only a few minor imperfections in it i.e. explanations of acronyms such as ROS, HMs and MDA are missing. De novo should be in italics 

Author Response

Dear reviewers,

Thank you very much for your kindly comments on our manuscript. We have carefully considered the suggestion of reviewer and make some changes. Please see the attachment file.

Reviewer 2 Report

The present review is very interesting as microRNA are key components for plant growth, development, and defence. I appreciated to read this text that is well-structured and written.

However, I have some comments for the authors to improve their work:

-the definition of miRNAs is not correct (see abstract for example): they are small non-coding RNA and not regulator of ribosomes. So, change the definition.

-part of the text is in red (keywords, references..), why?

-at the beginning of the work the authors should better clarify and underline  that microRNa are involved in different functions of plants, in response to both abiotic and biotic stress and during plant development. Moreover, they should also declare that recently microRNAs have been also documented to be able to promote the transcription of their target, not acting always as a negative regulator for them. I suggest to see and cite the following works: doi: 10.1155/2014/970607;  https://doi.org/10.1007/s00344-022-10686-2; https://doi.org/10.1007/s00344-022-10686-2; 

Author Response

Dear reviewer:

Thanks for your constructive comments. We have carefully considered the suggestions and done our best to revise this manuscript. Please see the attachment file.

This manuscript is a resubmission of an earlier submission. The following is a list of the peer review reports and author responses from that submission.

Round 1

Reviewer 1 Report

Dear Authors,

The manuscript is well written, with a few minor shortcomings that I will list below. It provides an interesting and entertaining read and will be a good source of "digested" information for many researchers dealing with heavy metals and their effects on plants. 

General comments:

The journal heading is missing.

Citations should be preceded by a space.

Gene names should always be written in italics.

Latin species names are three-part names. At first use, provide all three parts, then use the abbreviated name.

Introduce the term isomR and use it. 

Specific comments:

Line 54: change "pre-cursor" to "precursor"

Figure 1: resolution too low, in the caption give the name of the program and its citation. 

Table 1: citations are inconsistent with journal format and thus difficult to find in the literature list, correct formatting of gene names as noted above, correct Latin species names as noted above. S. Cultivars replace with Saccharum spp. 

Lines 136-138: add citations for A.thaliana and O.sativa.

Figures 2 and 3: resolution too low. Missing are citations for the papers on which the diagrams are based. 

Lines 286-289: add references. The sentence indicates plural "studies" and there is only one reference.

Line 324: change "Ck" to "CK"

The "Conclusions and perspectives" section should be shortened. It repeats information from the main body of the manuscript in many places.

Best,

M

Author Response

Thanks for reviewer’s kind comments on our manuscript. There is no doubt that these comments are valuable and very helpful for improving our manuscript. Here we submit the revision version of manuscript with the title “microRNAs: Key players in plant response to metal toxicity”, which has been modified according to the reviewers’ suggestions, and mark all the changes in red in the revised manuscript. In what follows, we would like to answer the questions reviewer mentioned and give detailed account of the changes made to the original manuscript.

Reviewer 2 Report

The reason for this decision is:

This manuscript does not fulfill the standards established for the journal to be considered for publication.

This review describes the toxicity of plants. But I still doubt whether to use the word toxic in plants. The reason is widely known that plants react. Of course, some Arabidopsis scientists use it, but it is a confusing data in academia. Using Table 1, the representation of Figure 1, Figure 2, and Figure 3 seems to indicate the result of an excessive delusion. In addition, it is necessary to re-examine whether the use of NGS is useful for our studies.

Author Response

Thanks very much for taking your time to review this manuscript. I really appreciate all your comments and suggestions! The following is our point-by-point response to the comments raised by the reviewers. Please see the attachment. 

Round 2

Reviewer 2 Report

The correlation between genes and metal toxicity is insufficient, but it does not explain the very broad meaning of a response related to plant growth as if it were an experimental result. In addition, various phytotoxic reactions and examples of various crops should be explained.